# The Pathogenesis of Adenomyosis vis-à-vis Endometriosis

**DOI:** 10.3390/jcm9020485

**Published:** 2020-02-10

**Authors:** Sun-Wei Guo

**Affiliations:** 1Shanghai Obstetrics and Gynecology Hospital, Fudan University, Shanghai 200011, China; hoxa10@outlook.com; 2Shanghai Key Laboratory of Female Reproductive Endocrine-Related Diseases, Shanghai 200011, China

**Keywords:** adenomyosis, endometrial-myometrial interface disruption, endometriosis, pathogenesis, pathophysiology, repeated tissue injury and repair

## Abstract

Adenomyosis is used to be called endometriosis interna, and deep endometriosis is now called adenomyosis externa. Thus, there is a question as to whether adenomyosis is simply endometriosis of the uterus, either from the perspective of pathogenesis or pathophysiology. In this manuscript, a comprehensive review was performed with a literature search using PubMed for all publications in English, related to adenomyosis and endometriosis, from inception to June 20, 2019. In addition, two prevailing theories, i.e., invagination—based on tissue injury and repair (TIAR) hypothesis—and metaplasia, on adenomyosis pathogenesis, are briefly overviewed and then critically scrutinized. Both theories have apparent limitations, i.e., difficulty in falsification, explaining existing data, and making useful predictions. Based on the current understanding of wound healing, a new hypothesis, called endometrial-myometrial interface disruption (EMID), is proposed to account for adenomyosis resulting from iatrogenic trauma to EMI. The EMID hypothesis not only highlights the more salient feature, i.e., hypoxia, at the wounding site, but also incorporates epithelial mesenchymal transition, recruitment of bone-marrow-derived stem cells, and enhanced survival and dissemination of endometrial cells dispersed and displaced due to iatrogenic procedures. More importantly, the EMID hypothesis predicts that the risk of adenomyosis can be reduced if certain perioperative interventions are performed. Consequently, from a pathogenic standpoint, adenomyosis is not simply endometriosis of the uterus, and, as such, may call for interventional procedures that are somewhat different from those for endometriosis to achieve the best results.

## 1. Introduction

Adenomyosis, defined as the presence of endometrial glands and stroma infiltrated deep and haphazardly into the myometrium [1], is a common gynecologic disorder with poorly understood pathogenesis and pathophysiology [2,3,4]. Its presenting symptoms include a soft and diffusely enlarged uterus with pelvic pain, abnormal uterine bleeding (AUB), and subfertility [5,6,7,8]. 

The definition of adenomyosis is essentially the presence of ectopic endometrium within the myometrium, very similar to that of endometriosis, which is the presence of ectopic endometrium outside of the uterine cavity. In other words, the major culprits of the two diseases, i.e., ectopic endometrium, are the same, but their locations are different. As such, adenomyosis is used to be called “endometriosis interna” [9], presumably due to such a similarity. 

The resemblance between adenomyosis and endometriosis appears to go beyond the similarity in their definition. Aside from their similar symptomologies, both are estrogen-dependent [10] and share many molecular aberrations. The clinical treatment strategy for the two diseases also seems to be similar, and the drugs to treat the two diseases can be used interchangeably [11]. In fact, few drugs are being developed exclusively for adenomyosis [12]. 

In addition, both diseases face similar challenges. Medical treatments of the two diseases are tenuous [12]. Both are badly in need of a better classification system [8,13,14]. Given the similarity between the two conditions, one question is: from the pathogenesis or pathophysiology standpoint, is adenomyosis simply endometriosis of the uterus? 

In this review, I shall provide an overview of the pathogenesis of the two conditions and underscore the difference in their genesis and the local microenvironments that shape the lesional destiny. In particular, I shall point out some loose ends in a prevailing theory on adenomyosis pathogenesis, and propose a new and alternative hypothesis. Hence, the short answer to the question would be: adenomyosis is not simply endometriosis of the uterus; rather, it shares with endometriosis similar developmental features but probably not the causes. 

## 2. Epidemiology 

The reported prevalence of adenomyosis varies greatly among different studies, from as low as 10% to as high as 57% [15,16]. This wide variability stems from variations in the histological criteria adopted for diagnosis and the techniques used to procure myometrial samples [15]. In a rare population-based study, the estimated prevalence of adenomyosis in Northeast Italy is reported to be 0.18%, much lower than that of endometriosis (1.14%) [17]. The age-specific incidence of adenomyosis is very different from that of endometriosis, reaching its peak at the age interval of 46–50 years [17]. These numbers seem to be low, possibly because the cases were all histologically confirmed. With the shift in diagnosis from hysterectomy to imaging due to advancing imaging technologies, it is certain that the incidence peak may change to women younger than 46–50 years [3]. 

Parity is consistently shown to be a risk factor, while smoking is inconsistently shown to be a protective factor [15,18]. A preponderance of evidence suggests that a history of receiving dilatation and curettage (D&C) procedures is a risk factor for adenomyosis [19,20,21,22]. The D&C procedure, and perhaps other uterine surgeries as well, may disrupt the endometrial–myometrial interface (EMI) and facilitate the invasion, implantation, embedding, and establishment of endometrial colonies within the myometrial wall, increasing the risk of adenomyosis. 

The risk profiles for adenomyosis are thus quite different from that of endometriosis, in which greater parity is a protective factor, while earlier age at menarche and shorter menstrual cycle length—factors that are consistent with Sampson’s retrograde menstruation theory for the etiology of endometriosis—are risk factors [23]. Even in women with leiomyomas, women with both leiomyomas and adenomyosis are reported to be older than those with leiomyomas alone [24]. The different risk profiles and the different age-specific incidence between the two diseases indicate that they are likely to have different pathogenesis. 

## 3. Pathogenesis

For endometriosis, several theories regarding its pathogenesis have been proposed, and these hypotheses can be divided into three themes: in situ development (such as coelomic metaplasia or embryonic cell rests), implantation, or a combination of in situ development and implantation [25,26]. Sampson’s retrograde menstruation theory is the most widely accepted and has the most robust evidence [27]. 

In sharp contrast to endometriosis, the exact pathogenesis of adenomyosis is largely a conundrum. Currently, there are essentially two major prevailing and competing theories: invagination and metaplasia [4]. In a recent expert review, García-Solares et al. summarized these two theories [4]. The invagination theory was based largely on the tissue injury and repair (TIAR) theory proposed by Leyendecker and his colleagues [28,29,30]. The TIAR theory also attempts to account for the pathogenesis of endometriosis. Regardless, even within the TIAR framework it seems evident that the pathogenesis of adenomyosis differs from that of endometriosis [28,29,30]. 

### 3.1. Invagination

The TIAR theory is based on several keen observations. First, the uterus is composed of two phylogenetically and ontogenetically different organs called archimetra and neometra [31]. The former consists of endometrium, the subendometrial unit called the junctional zone (JZ) [32], and the underlying stratum subvasculare of the myometrium. It is the oldest part of the uterus, originated paramesonephrically, and displays a cyclic pattern of steroid hormone receptor expression. The neometra, in contrast, is of non-Müllerian origin and exhibits a constantly high receptor expression throughout the cycle [31]. Of note, the archimetra and the neometra function differently during the process of reproduction [31].

Remarkably, the change in uterine zonal anatomy depends on ovarian sex steroid hormones [33,34]. In particular, the JZ is involved in the modulation of uterine contractions throughout the menstrual cycle, ostensibly serving the purpose of sperm transport during ovulation and antegrade menstruation during the follicular and menstrual phases [35,36]. Thus, the changes in JZ morphology, such as an irregular increase in thickness, has been postulated to be the basis for MRI diagnosis of adenomyosis [37]. However, the JZ thickening, in and of itself, does not provide proof of mucosal invasion of the myometrium [37]. 

Second, uterine peristalsis plays important roles in reproduction, a finding first reported in the early 1990s [35,36] and later characterized further by Dr. Leyendecker’s group [38,39,40,41]. In a nutshell, they showed that uterine peristalsis is critical for sperm transport, and women with endometriosis exhibit hyperperistalsis and dysperistalsis [38,39,40,41]. It is well accepted that uterine peristalsis due to myometrial contraction originates from the JZ, while the outer myometrium remains quiescent [42]. 

Uterine peristalsis can be induced by oxytocin (OT) [43], which is mediated by its receptor OTR. In myometrium, OT production and OTR expression seem to be regulated by estrogen receptor α (ERα) [44,45]. In adenomyosis, OTR overexpression in myometrium, concomitant with uterine hyperactivity, has been reported and is found to correlate to the severity of dysmenorrhea [46]. 

Third, uterine hyperperistalsis/dysperistalsis may be a result of archimetral hyperestrogenism [38,47]. In addition, genes involved in estrogen biosynthesis, such as steroidogenic acute regulatory protein (StAR) and P450 aromatase, are upregulated in adenomyosis [48,49]. Moreover, there is a remarkable similarity between hyperestrogenism in adenomyosis and the role of estrogen in wound repair [29,30]. 

Lastly, many genes, such as ERβ and cyclooxygenase-2 (COX-2), that are involved in tissue repair and inflammation, are upregulated in adenomyosis. Thus, a basic molecular pathway diagram underlying the TIAR theory can be depicted as in Figure 1: tissue injury in the JZ results in inflammation and, in particular, elevated IL-1β expression, which can induce COX-2, the gene coding for the rate-limiting enzyme for the production of prostaglandin E2 (PGE_2_), which, in turn, can upregulate P450 aromatase. The overexpression of aromatase, in conjunction with the upregulation of StAR, can increase the production of estradiol, which induces ERβ overexpression, causing ensuing changes in a feed-forward fashion. 

The TIAR theory postulates that repeated and sustained overstretching due to hyperperistalsis would cause injury of the myocytes and fibroblasts in the JZ close to the fundo-cornual raphe, an event termed microtrauma [29,30]. The microtrauma would focally activate the TIAR system with increased inflammation and local production of estradiol, which, in turn, induce more inflammation and more estrogen production, establishing a feed-forward loop that further induces uterine hyperperistalsis through ERα induction of the OT/OTR system. The chronic hyperperistalsis in the JZ facilitates repetitive autotraumatization, causing the disruption of the muscular fibers in the EMI, eventually leading to the invagination of the endometrial basal layer into the myometrium, and thus adenomyosis. Following [4], Figure 2 depicts the core component of the TIAR theory. 

### 3.2. Metaplasia

The metaplasia theory proposes that adenomyotic lesions may originate from the metaplasia of displaced embryonic pluripotent Müllerian remnants [50]. Alternatively, endometrial stem/progenitor cells or their niche cells may behave aberrantly for some unknown reason, and their differentiating progeny cells move towards the myometrium, rather than *functionalis*, resulting in adenomyosis [51]. 

### 3.3. The TIAR Theory: The Need for Patching up

Both invagination and metaplasia theories are inspiring and somewhat thought-provoking, and help us to understand some basic features of adenomyosis. Indeed, the invagination theory may be consistent with the epidemiological findings that multiparity and uterine surgery are risk factors for adenomyosis due to their potential to disrupt the JZ. The metaplasia theory could account for some cases of adenomyosis in the rudimentary muscular uterine wall of patients with Mayer–Rokitansky–Kuster–Hauser syndrome [4]. 

Yet a good theory should satisfy at least three basic requirements: (1) it is falsifiable; (2) it can explain existing data/phenomena; and (3) it can make useful predications. There may be some additional requirements, but, among them, biological plausibility is also of importance. While both theories may be biologically plausible, neither theory, unfortunately, could fully explain the genesis of different subtypes of adenomyosis. More importantly, it is difficult, if not impossible, to devise an experiment to prove or disprove either theory. It is also challenging to use either theory to make some useful, previously unknown, predictions.

In fact, many questions can be raised regarding the two theories, particularly the TIAR theory. Is there any trigger, if any, that leads to the metaplasia, or the microtraumatization, that eventually results in invagination? If microtraumatization is ubiquitous, why do only a small portion of women eventually develop adenomyosis? Can we prevent the microtrauma, or at least tame it, so that the risk of developing adenomyosis can be substantially reduced? In the development of adenomyosis, epithelial–mesenchymal transition (EMT) is thought to play a role, but how does the TIAR theory accommodate it? As stem cells are thought to be involved in adenomyosis [52], how can the TIAR theory account for them?

To be fair, a single theory may not be able to account for the pathogenesis of all subtypes of adenomyosis. For example, prolonged and unopposed exposure to estrogens is found to induce adenomyosis in various animals [53,54,55,56,57]. In particular, neonatal exposure to diethylstilbestrol followed by continuous hormonal treatment—even progesterone alone—also results in adenomyosis [58]. In addition, the grafting of a single anterior pituitary into the uterine lumen induced a high incidence of adenomyosis in some strains of mice [59], but inhibition of pituitary prolactin secretion during youth in mice suppressed the induction of adenomyosis [60], suggesting the potential role of prolactin in the pathogenesis of adenomyosis. 

In view of the above, it seems that a one-size-fits-all theory may be too idealistic, too ambitious, and simply unattainable. Indeed, the newly reported association between focal adenomyosis of the outer myometrium (FAOM) and deep endometriosis (DE) [61,62] suggests that not all subtypes of adenomyosis are the same and may have a different pathogenesis. 

Close scrutiny of the original TIAR theory also raises some uncertainty and questions. In this theory, the first, critical, and initiating event is the microtraumatization. That is, “…repeated and sustained overstretching and injury of the myocytes and fibroblasts at the endometrial-myometrial interface close to the fundo-cornual raphe would activate the TIAR system focally with increased local production of estradiol”, constituting the first-step injury that subsequently leads to the activation of the TIAR system and the auto-traumatization by hyperperistalsis [29,30]. 

Here, the “primum movens” that sets off the ensuing chain events leading to the formation of adenomyotic lesion is the mechanical stretching due to uterine peristalsis, which presumably activates COX-2 and the subsequent increased production of PGE_2_ [63], which, in turn, results in increased local production of estrogen in normal endometrial stromal cells (Figure 1). 

However, the feed-forward loop, shown in Figure 1, that underpins the TIAR theory, is based on findings almost exclusively from *endometriotic* stromal cells. In fact, the feed-forward loop depicted in Figure 1 is embedded within the positive feedback model depicted in Figure 11 in [64]. Yet *normal* endometrial stromal cells are very different in behavior from endometriotic or even adenomyotic stromal cells. In fact, over 1000 genes are documented to be differentially expressed between adenomyotic lesions and eutopic endometrium [65] and close to 1000 genes are differentially expressed between ectopic and eutopic endometrium [66]. As such, the difference between ectopic and normal endometrium would be conceivably no less dissimilar. In stark contrast to endometriotic stromal cells, in fact, the promoter of the gene coding for P450 aromatase is hypermethylated in *normal* endometrial stromal cells, and thus silenced [67,68]. Similarly, the gene encoding for steroidogenic factor-1 (SF-1), the protein critically involved in estrogen biosynthesis, is also hypermethylated and silenced in *normal* endometrial stromal cells [69]. Even when stimulated with PGE_2_, the expression of most genes involved in estrogen biosynthesis, including StAR, SF-1, and aromatase, are still negligibly low, concomitant with the very low production of estradiol in *normal* endometrial stromal cells [70]. In addition, ERβ is also hypermethylated and silenced in *normal* endometrial stromal cells [71]. Thus, hyperperistalsis may induce COX-2 expression and higher PGE_2_ production in endometrial stromal cells, but it is unlikely to further induce increased estrogen production. These facts cast some shadows on the very foundation of the TIAR theory. 

### 3.4. Revamping the TIAR Theory: The Endometrial-Myometrial Interface Disruption (EMID)

It is clear from the above discussion that microtraumatization-induced estrogen production may not occur in the first place. Hence, the very foundation of the TIAR theory seems to be in doubt. Can the theory be salvaged? Is there such a key event in the initial stage of the genesis? If yes, what is it?

Our knowledge of the molecular mechanisms underlying TIAR or wound healing has been expanded greatly in the last two decades. Tissue injury causes a disruption of local vasculature and extravasation of blood, leading to platelet aggregation and the formation of clots. Vascular damage results in the loss of perfusion and consequent hypoxia. The tissue hypoxia can be further exacerbated by an influx of inflammatory and stromal cells—all with high metabolic demands for oxygen, which is essential for all aerobic organisms to produce energy via mitochondrial oxidative respiration and to perform other vital biological functions [72]. Once hypoxic, evolutionarily conserved adaptive processes, such as hypoxia-inducible factors (HIFs), are activated [73]. Thus, hypoxia may well be viewed as the single most important event following tissue injury, and, indeed, adaptation to hypoxic stress via HIF signaling plays a critical role in promoting wound healing processes [74,75]. 

In iatrogenic trauma such as the D&C procedure, the injury to the endometrium can probably extend to the EMI. Hence, the injury to the EMI constitutes a trauma, leading to subsequent tissue hypoxia in myocytes and fibroblasts. One hallmark of hypoxia is the overexpression of the transcription factor HIF-1α, a key mediator of cellular adaptation to low oxygen levels. In response to hypoxia, macrophages recruited to the wounding site release chemotactic factors and secrete growth factors vital for cell migration and proliferation that facilitate tissue repair. Macrophage-secreted growth factors such as vascular endothelial growth factor (VEGF) and its receptor VEGFR1 and VEGFR2, platelet-derived growth factor (PDGF), fibroblast growth factor (FGF), and transforming growth factor-β (TGF-β) have been reported to be induced by hypoxia [76,77,78]. Consequent to vascular damage, activated platelets can also release an array of cellular growth factors and angiogenic factors such as PDGF and VEGF, as well as inflammatory mediators such as IL-8 [79,80,81]. 

The activation of HIF-1α and the formation of the HIF complex induce several angiogenic growth factor genes, including VEGF and stromal cell-derived factor-1 (SDF-1)/C-X-C motif chemokine 12 (CXCL12) [82,83,84]. As tissue repair requires new blood vessel growth via angiogenesis and vasculogenesis, hypoxia-induced angiogenesis re-establishes vasculature within the injured tissues, facilitating their capacity to heal. 

Hypoxia can also enhance cellular viability and proliferation through upregulation of insulin growth factor 2 (IGF-2) and IGF binding protein 3 (IGF-BP3) [85]. In addition, hypoxia also induces COX-2 [86]. HIF-1α-induced COX-2 expression leads to elevated PGE_2_ levels, which can induce PGE_2_-mediated vascularization [87,88].

In addition, hypoxia-induced SDF-1/CXCL12 can attract hematopoietic and endothelial progenitor cells (EPCs) expressing its receptor CXCR4, facilitating vasculogenesis [89]. HIF-1α activation also plays a role in the homing of circulating EPCs to injured tissues experiencing hypoxia to facilitate the restoration of damaged vasculature. Incidentally or not, SDF-1 plays a critical role in recruiting bone marrow derived stem cells (BMDSCs) to endometrium and ectopic endometrium [90,91]. Similarly, platelets stimulate the homing of BMDSCs, facilitating angiogenesis in injured and hypoxic tissues [92].

Hypoxia can also induce EMT [93,94]. In endometrial epithelial cells, the hypoxia-induced EMT is facilitated by enhanced autophagy [95]. Thus, EMID may induce EMT in adjacent endometrial epithelial cells and enhance their mobility and invasiveness, resulting in the crossing of EMI to the myometrium by endometrial epithelial cells. This would effectively establish a viable colony of ectopic endometrium, or, at the very least, gain a crucial foothold in establishing an adenomyotic lesion. 

Remarkably, hypoxic conditions alone can dramatically change the behavior of endometrial cells that are otherwise “normal”, manifesting in increased estrogen production and other “abnormal” phenotypes [96]. More remarkably, activated platelets can activate HIF-1α in endometrial and endometriotic stromal cells, effectively generating a state of hypoxia [97]. In fact, activated platelets can also upregulate the gene expression of StAR, SF-1 and aromatase, increasing the estrogen production significantly in endometriotic stromal cells [98]. Thus, the increased estrogen levels may induce an ERα-mediated OT/OTR system, enhancing uterine peristalsis. 

Note that hypoxia-induced COX-2 overexpression can increase the production of not only PGE_2_ but also PGH_2_, which leads to increased production of thromboxane A2 (TXA_2_), a vasoconstrictor, a potent platelet activator, and a potent inducer of uterine contraction [99]. Hence hypoxia-induced COX-2 expression would also lead to increased uterine peristalsis, effectively facilitating the perpetuation of the vicious cycle. 

As a tissue trauma, EMID may also activate the hypothalamic–pituitary–adrenal (HPA) axis, causing the release of catecholamines, especially adrenaline/noradrenaline. It also may induce the release of PGE_2_. Together, these catecholamines and PGE_2_ would impair the cell-mediated immunity, promoting the survival of displaced and dispersed cells that spread to the EMI and possibly beyond due to the procedure. 

Thus, when EMID occurs, such as in the D&C procedure, the resultant tissue hypoxia would lead to platelet aggregation, increased estrogen production, induction of the TGF-β1, VEGF, PDGF, COX-2, and SDF-1 signaling and the resultant elevated PGE_2_ production, induction of EMT, and increased recruitment of BMDSCs in the wound site, all of which can promote the mobility and implantation of endometrial cells to the myometrium. In addition, increased estradiol levels would induce ERα, which further activate OT/OTR signaling pathway, causing increased peristalsis. The release of adrenaline/noradrenaline and PGE_2_ can reduce cell-mediated immunity, promoting the survival of spread and displaced endometrial cells. 

This hypothesis of adenomyosis pathogenesis can be schematically depicted in Figure 3. This model can at least account for the genesis of adenomyosis arising from iatrogenic trauma such as uterine procedures, but perhaps not for adenomyosis induced by other causes. Thus, this hypothesis may have its limitations. For a disease with a poorly understood pathogenesis, however, focusing on something that is fairly common and well supported by epidemiological data, which could perhaps be modifiable might be a good start. Aside from understanding of the pathogenesis for one class of adenomyosis, the knowledge thus gained may illuminate the pathogenesis of adenomyosis due to other causes.

The model depicted in Figure 3 is completely falsifiable. In fact, EMID can robustly induce adenomyosis in mice (Hao et al., submitted for publication). The model also seems to be consistent with published data, i.e., an increased risk of adenomyosis in women with a history of D&C or in multiparous women [19,20,21,22]. It overcomes several deficiencies of the TIAR theory by accommodating EMT and the recruitment of BMDSCs. It provides explanations as to why hyperperistalsis occurs. 

Most importantly, the model can make several predictions. First, since iatrogenic procedures such as a D&C procedure may lead to the release of catecholamines and PGE_2,_ which impair the cell-mediated immunity that promotes the survival of endometrial cells dispersed and displaced by the procedure, it raises a tantalizing possibility that perioperative intervention could reduce the risk of adenomyosis, as in the case of a reduction in recurrence risk in endometriosis [100]. In addition, since uterine procedures using different energy sources result in different severities of injury and bleeding patterns, the resultant EMID would lead to differential = risk of developing adenomyosis. For example, the use of coagulation might lead to a lower risk of adenomyosis than the D&C procedure. 

To distinguish from the TIAR theory, this model, depicted in Figure 3, is called **e**ndometrial–**m**yometrial **i**nterface **d**isruption (EMID) hypothesis. While it can explain the genesis of adenomyosis induced by iatrogenic trauma, it may also explain the pathogenesis of adenomyosis due to persistent hyperperistalsis that causes EMID, but probably not FAOM or extrinsic/exterior adenomyosis. 

Several cautions are in order. First, there may well be other aiders and abettors, yet to be identified, that also play roles in the genesis of adenomyosis. Second, this model is by no means final or complete, and is subject to change or modification as we further elucidate the underlying molecular mechanisms of adenomyosis pathogenesis. Lastly, to account for hyperperistalsis-induced EMID, the upstream cues or risk factors need to be elucidated. 

### 3.5. Similarities and Difference in Pathogenesis between Adenomyosis and Endometriosis

From the above discussion, it is clear that adenomyosis and endometriosis have a different pathogenesis. In fact, even for adenomyosis, there might be two or more different pathogenic origins, for example, the FAOM, or extrinsic or external adenomyosis, might originate from the invasion of neighboring DE lesions. Future phylogenic analysis based on the next-generation sequencing should be able to resolve this issue [101]. Even within the framework of the TIAR theory, endometriosis also originates from the microtraumatization, and the pathogenesis seems to be different from that of adenomyosis. One probable exception might be the abdominal wall endometriosis (AWE), which is associated with cesarean delivery and abdominal hysterectomy [102,103]. Conceivably, these surgical procedures would result in the direct implantation of endometrial cells ectopically within the abdominal wall. 

## 4. Conclusions 

In this review, after pointing out some deficiencies in the TIAR theory, the EMID hypothesis has been proposed in order to revamp the theory to account for the pathogenesis of adenomyosis due mostly to iatrogenic procedures. Unlike the TIAR theory, the EMID hypothesis is easily falsifiable, at least in mouse. The EMID hypothesis is also consistent with epidemiological data stating that iatrogenic uterine procedures such as induced abortion and D&C are a risk factor for adenomyosis. More importantly, the hypothesis raises the prospect that perioperative interventional procedures could be instituted to reduce or minimize the risk of adenomyosis resulting from iatrogenic procedures aplenty nowadays in gynecological practice. It is evident that, even within the TIAR or the EMID framework, adenomyosis has a different pathogenesis than that of endometriosis. In this sense, adenomyosis is *not* simply endometriosis of the uterus. 

For adenomyosis pathophysiology, the hypothesis of repeated tissue injury and repair (ReTIAR) has been proposed [104]. As adenomyosis shares with endometriosis in cyclic bleeding and thus ReTIAR, the two diseases also share common key molecular events in lesional development, i.e., EMT, fibroblast-to-myofibroblast transdifferentation, smooth muscle metaplasia, and fibrogenesis [105,106]. While these key molecular events are similar or even identical for the two diseases, their respective lesional microenvironments are not entirely the same. Notably, the proximity to myometrium means that adenomyotic lesions may experience stronger and more frequent contractions, and thus more mechanical stretch and strain, especially when the uterus is enlarged with thickening JZ, which may accelerate the tempo and pace of lesional fibrogenesis. Even though adenomyotic lesions (used to be called endometriosis interna) may resemble DE lesions (now called adenomyosis externa [107]) in their look and feel, their developmental paths may be similar but not entirely the same, due to their different microenvironment. Consequently, and again, adenomyosis is *not* simply endometriosis of the uterus, at least from the pathogenesis perspective. 

The similarity in the natural history between endometriosis and adenomyosis may explain why many medications used primarily for treating endometriosis can and have been used also for treating adenomyosis. However, there are still some notable differences in microenvironment between the two conditions, as discussed above. Therefore, therapeutics that capitalize on these differences may stand a better chance to be more efficacious for adenomyosis per se. For example, a compound that arrests or even reverses the lesional development processes would be promising and thus admissible, but if the compound can additionally suppress uterine hyperperistalsis should stand a better chance to be efficacious in managing adenomyosis. 

That adenomyosis is not simply endometriosis of the uterus implies that, when it comes to management and prevention, interventions that are carefully tailored to adenomyosis should be more effective. For example, perioperative intervention when iatrogenic procedures such as D&C are performed may reduce the risk of developing adenomyosis in the first place. However, there are still many unknowns. Future research is warranted to elucidate the molecular mechanisms underlying the EMID and ReTIAR, and the upstream cues or risk factors that led to EMID-inducing hyperperistalsis. As of now, at least we have something specific and tangible to work on, and the insight we will gain from the mechanisms of EMID-induced adenomyosis should hopefully help us better understand the pathogenesis of adenomyosis. 

## Figures and Tables

**Figure 1 jcm-09-00485-f001:**
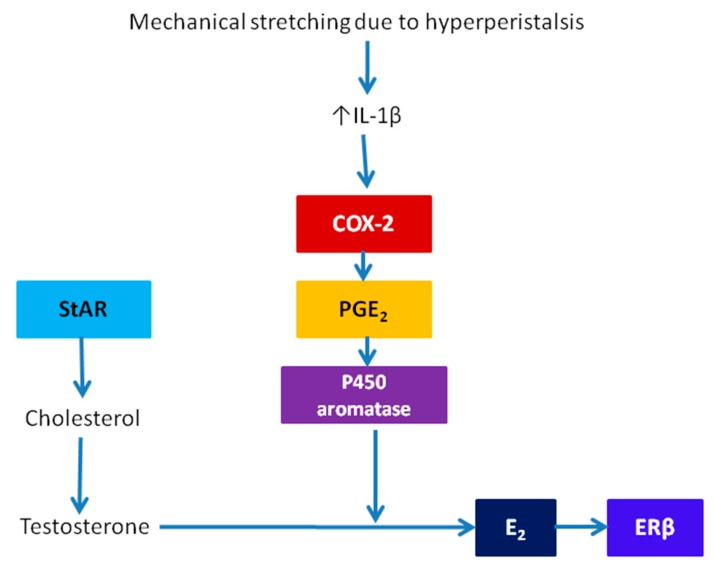
The key molecular signaling events initiated by the tissue injury and repair (TIAR) that leads to the increased local production of estradiol, as proposed by Leyendecker et al. [29,30]. Gene/protein names: COX-2: cyclooxygenase-2; E_2_: 17β-estradial; ERβ: estrogen receptor β; IL-1β: interleukin-1β; P450 aromatase: aromatase; PGE_2_: prostaglandin E2; StAR: steroidogenic acute regulatory protein.

**Figure 2 jcm-09-00485-f002:**
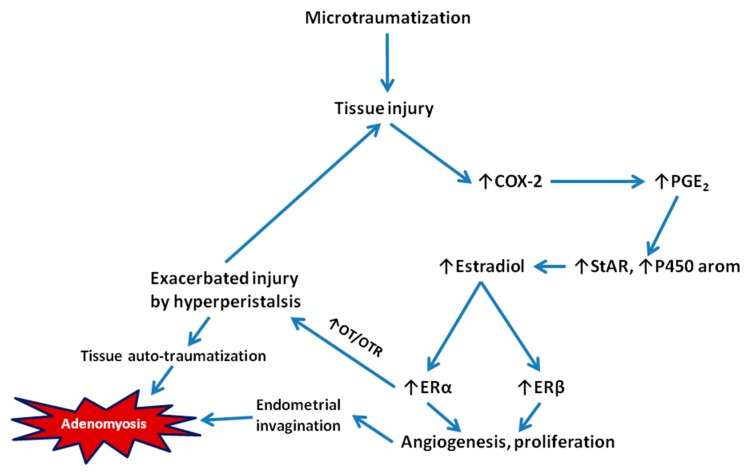
Leyendecker’s model of tissue injury and repair (TIAR) that initiates the genesis of adenomyotic lesions [29,30]. Briefly, microtraumatization in the endometrial–myometrial interface causes tissue injury, which subsequently induces upregulation of COX-2 and increased production of PGE_2_, which, in turn, induces the expression of genes critical to estrogen production such as StAR and aromatase, resulting in increased local estrogen production. The elevated estrogen levels would activate both ERα and ERβ, leading to the induction of the OT/OTR signaling and subsequent increased uterine peristalsis and increased angiogenesis and proliferation. The increased peristalsis would further exacerbate uterine hyperperistalsis and thus TIAR, causing endometrial invagination and ultimately the formation of adenomyotic lesions. Gene/protein names: COX-2: cyclooxygenase-2; E_2_: 17β-estradial; ERα: estrogen receptor α; ERβ: estrogen receptor β; IL-1β: interleukin-1β; P450 aromatase: aromatase; OT: oxytocin; OTR: oxytocin receptor; PGE_2_: prostaglandin E2; StAR: steroidogenic acute regulatory protein.

**Figure 3 jcm-09-00485-f003:**
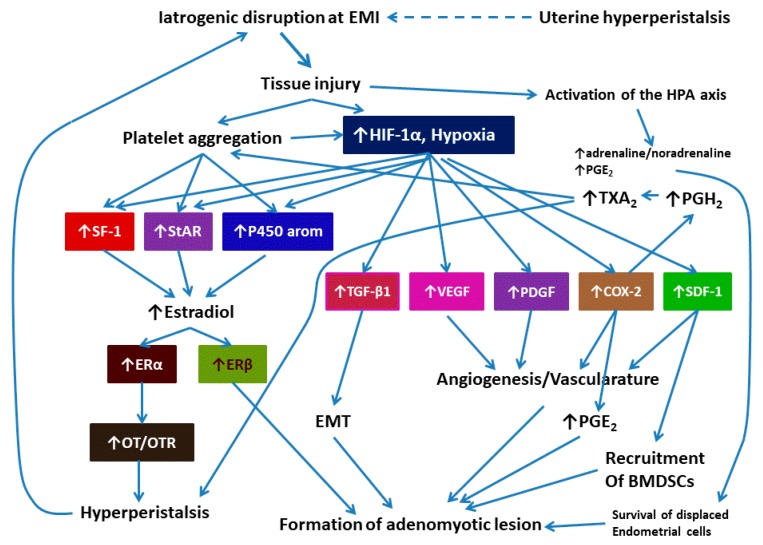
Schematic illustration of the formation of adenomyotic lesions due to the endometrial–myometrial interface disruption (EMID). Iatrogenic procedures causes disruption at the endometrial–myometrial interface (EMI), which leads to platelet aggregation and the induction of HIF-1α, effectively causing tissue hypoxia. Uterine hyperperistalsis may also induce EMI disruption (shown in dashed arrow). As a result, genes involved in estrogen production are upregulated, resulting in increased local production of estrogen and subsequent induction of both ERα and ERβ, which, in turn, leads to the induction of the OT/OTR signaling and increased uterine peristalsis. In addition, tissue hypoxia activates TGF-β1, VEGF, PDGF, COX-2, and SDF-1 signaling pathways, leading to increased angiogenesis, vascularature, and the recruitment of BMDSCs to the wounding site. The induction of COX-2 would also increase the production of PGH_2_ and TXA_2_, which also enhances uterine peristalsis. Moreover, the TGF-β1 signaling pathway induces EMT, leading to the invasion of endometrial epithelial cells to the EMI and further down to the myometrium. Tissue injury also would activate the HPA axis, leading to the release of catecholamines and PGE_2_, which collectively result in impaired cell-mediated immunity and, as such, enhances the survival of displaced and dispersed endometrial cells within the myometrium. All these events ultimately lead to the formation of adenomyotic lesions in the myometrium. Abbreviations used: BMDSC: bone marrow derived stem cells; COX-2: cyclooxygenase-2; E_2_: 17β-estradial; EMI: endometrial-myometrial interface; EMT: epithelial-mesenchymal transition; ERα: estrogen receptor α; ERβ: estrogen receptor β; HIF-1α: hypoxia-inducible factor 1α; HPA: hypothalamic-pituitary-adrenal; P450 aromatase: aromatase; PDGF: platelet-derived growth factor; OT: oxytocin; OTR: oxytocin receptor; PGE_2_: prostaglandin E2; PGH_2_: prostaglandin H2; SDF-1: stromal cell-derived factor 1; SF-1: steroidogenic factor-1; StAR: steroidogenic acute regulatory protein; TGF-β1: transforming growth factor β1; TXA_2_: thromboxane A2; VEGF: vascular endothelial growth factor.

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
