# Peer review of "The Pathogenesis of Adenomyosis vis-à-vis Endometriosis"

_jcm, 2020, doi:10.3390/jcm9020485_

Round 1

Reviewer 1 Report

This is a rather comprehensive review of the pathophysiology of adenomyosis in comparison with endometriosis.

Typically, when the title is in the form of a closed question (a yes or no question) the reader assumes an affirmative answer from the article "a yes".  The author's view is that adenomyosis is in fact NOT simply endometriosis of the uterus.  Therefore, I suggest changing the title to a statement rather than a question "Adenomyosis is not simply endometriosis of the uterus" The text is too long, in the abstract and in the body of the article.  I suggest reducing the article by cutting some of the molecular detailed content. The article will be strengthened by including some photomicrographs of examples from both entities

Author Response

This is a rather comprehensive review of the pathophysiology of adenomyosis in comparison with endometriosis.

I would like to thank the reviewer for his/her careful reading and constructive comments.

Typically, when the title is in the form of a closed question (a yes or no question) the reader assumes an affirmative answer from the article "a yes".  The author's view is that adenomyosis is in fact NOT simply endometriosis of the uterus.  Therefore, I suggest changing the title to a statement rather than a question "Adenomyosis is not simply endometriosis of the uterus" The text is too long, in the abstract and in the body of the article.  I suggest reducing the article by cutting some of the molecular detailed content. The article will be strengthened by including some photomicrographs of examples from both entities 

The Reviewer has an excellent point. In this revision, I have changed the title to “The pathogenesis of adenomyosis vis-à-vis endometriosis”. This may be more appropriate, since the main focus of the manuscript is on the pathogenesis hypothesis of adenomyosis. I also have revised the Discussion section. Many thanks.

In addition, I also have changed the phrase, “endometrial-myometrial interface trauma (EMIT)” to “endometrial-myometrial interface disruption (EMID)”. This is because in the process of preparing for the manuscript reporting the data that describe the successful establishment of a mouse model of EMID-induced adenomyosis (two mouse strains, and mechanically and thermally induced EMID), I realized that “trauma” is not well-defined, and to show “trauma” is thus a bit challenging. In contrast, to show “disruption” is straightforward. I hope this change is agreeable to the reviewer.

Reviewer 2 Report

The review article by Guo deals with a very interesting topic, regarding the definition of adenomyosis and the investigation of its resemblance with endometriosis, apart from the similar symptoms appearing in both conditions. It is a very comprehensive article, performed with a very detailed PubMed search, focusing initially on the limitations of the two prevailing theories regarding the pathogenesis of adenomyosis, the one referred to invagination and a second one referred to metaplasia. Furthermore, this review article analyzes the current understanding of wound healing, a new hypothesis called endometrial-myometrial interface trauma (EMIT), which has been proposed to account for adenomyosis resulting from iatrogenic trauma to EMI.  It provides an overview on the pathogenesis of the two conditions and underscores the difference in their genesis and local microenvironments that shape the lesional destiny. This manuscript has much information that could be very useful for readers. The studies cited in this review have been systematically put forward and the outcome of all the studies has been well summarized. Moreover, scientifically correct conclusions have been drawn when combining various independent studies to make a collective scientific statement. Overall, the review is a good read which provides a critical summary of the topic discussed. A weakness that can be pointed out refers to the lack of any genetic data that could differentiate endometriosis from adenomyosis.

Author Response

The review article by Guo deals with a very interesting topic, regarding the definition of adenomyosis and the investigation of its resemblance with endometriosis, apart from the similar symptoms appearing in both conditions. It is a very comprehensive article, performed with a very detailed PubMed search, focusing initially on the limitations of the two prevailing theories regarding the pathogenesis of adenomyosis, the one referred to invagination and a second one referred to metaplasia. Furthermore, this review article analyzes the current understanding of wound healing, a new hypothesis called endometrial-myometrial interface trauma (EMIT), which has been proposed to account for adenomyosis resulting from iatrogenic trauma to EMI.  It provides an overview on the pathogenesis of the two conditions and underscores the difference in their genesis and local microenvironments that shape the lesional destiny. This manuscript has much information that could be very useful for readers. The studies cited in this review have been systematically put forward and the outcome of all the studies has been well summarized. Moreover, scientifically correct conclusions have been drawn when combining various independent studies to make a collective scientific statement. Overall, the review is a good read which provides a critical summary of the topic discussed. A weakness that can be pointed out refers to the lack of any genetic data that could differentiate endometriosis from adenomyosis.

I would like to thank the reviewer for his/her careful reading, encouragement and constructive comments.

The reviewer is absolutely correct in saying that the manuscript lacks genetic data that could differentiate endometriosis from adenomyosis. This is mainly because that such an aspect has, to my best knowledge, never been paid any attention.

In addition, I also have changed the phrase, “endometrial-myometrial interface trauma (EMIT)” to “endometrial-myometrial interface disruption (EMID)”. This is because in the process of preparing for the manuscript reporting the data that describe the successful establishment of a mouse model of EMID-induced adenomyosis (two mouse strains, and mechanically and thermally induced EMID), I realized that “trauma” is not well-defined, and to show “trauma” is thus a bit challenging. In contrast, to show “disruption” is straightforward. I hope this change is agreeable to the reviewer.